# Analysis of the Direct Medical Costs of Colorectal Cancer in Antigua and Barbuda: A Prevalence-Based Cost-of-Illness Study

**DOI:** 10.3390/ijerph22040552

**Published:** 2025-04-03

**Authors:** Andre A. N. Bovell, Jabulani Ncayiyana, Themba G. Ginindza

**Affiliations:** 1Discipline of Public Health Medicine, School of Nursing and Public Health, University of KwaZulu-Natal, Durban 4000, South Africa; ncayiyanaj@ukzn.ac.za (J.N.); ginindza@ukzn.ac.za (T.G.G.); 2Cancer & Infectious Diseases Epidemiology Research Unit (CIDERU), College of Health Sciences, University of KwaZulu-Natal, Durban 4000, South Africa

**Keywords:** Antigua and Barbuda, colorectal, cost-of-illness, economic burden, colon, rectum, cost analysis, cancer

## Abstract

Colorectal cancer burden is a threat to health systems in several countries. As the cost of diagnosing, treating, and managing this cancer is unknown in Antigua and Barbuda, this study aimed to estimate its direct medical cost in this country. We used the prevalence-based cost-of-illness methodology to study data on patients diagnosed with colorectal cancer between 2017 and 2021. Data record abstraction was carried out to determine the five-year prevalence, and a top-down and bottom-up approach was employed to estimate the direct medical costs for colorectal cancer care components. All costs were computed at 2021 price levels and are reported in United States dollars. The total annual direct medical costs for colorectal cancer were estimated at USD 1.14 million (ranging between USD 0.85 million and USD 1.42 million). Major cost drivers were treatment (USD 613,650.01) and post-treatment side-effects care (USD 402,234.50). The overall estimated direct medical unit costs were USD 139,295.58, with the main drivers being surgery (USD 43,467.10), other complications of treatment (USD 28,469.21), and immunotherapy (USD 19,200.00). This study provides evidence of the economic burden of colorectal cancer in Antigua and Barbuda. The estimates of annual direct medical costs are substantial. Our findings could help in the development of health policy and aid in resource allocation related to local colorectal cancer management.

## 1. Introduction

Colorectal cancer is one of the leading cancers affecting populations globally [1,2], being the second and third most common cancer in women and men, respectively [1]. In 2022, there were an estimated 1.9 million new cases and 904,000 deaths from colorectal cancer [1], making it responsible for 9.6% of all diagnosed cancers worldwide in that year and representing a 0.2% increase from 2020’s estimate of 9.4% [1,2].

Known for its diverse risk factors, the chance of developing colorectal cancer increases markedly in patients older than 50 [3]. Notwithstanding this, recent evidence suggests a steadily rising and alarming incidence and/or increasing burden of early-onset colorectal cancer (disease in patients younger than 50 years) in several countries, especially within the last three decades [1,4]. This makes screening from as early as 45 to 50 years an important part of care in most developed and some developing countries [3,4]. Surgery is the only possible cure for most patients with colorectal cancer [5]. Additionally, postoperative chemotherapy is important in cases where the disease spreads [6]. For rectal cancers, preoperative radiotherapy plus chemotherapy is usually required [6], with surgery used for metastatic removal [6].

The stage at presentation has implications for overall disease management [7], with efforts to improve the detection and treatment of this cancer being contributors to the increase in the economic burden associated with this disease [8].

Several studies have estimated the economic burden of colorectal cancer using various cost-of-illness approaches, with most adopting either incidence- or prevalence-based methods [9,10]. More specifically, several researchers have opted to estimate the direct medical costs of colorectal cancer in various settings using the prevalence-based cost-of-illness methodology [9]. For example, Byun et al. estimated that its economic burden in Korea was KRW 3.1 trillion (Korean won), with direct costs estimated at KRW 1.97 trillion [11]. Vahdatimanesh et al. assessed this as USD 298,148.718, with direct medical costs accounting for 32.14% of total costs in Iran [12]. In Brazil, Rezende et al. found this to be Int 212 million (international dollars), with direct costs specific to colon cancer accounting for Int 134 million [13].

The prevalence-based methodology is commonly considered in costing studies due to its advantage in computing cost estimates attributable to a disease over a specified period, usually a year [10]. This can be achieved either retrospectively or prospectively, with key drawbacks being that it does not quantify the long-term consequences of the condition being measured, nor does it measure potential savings resulting from certain health interventions [10,14].

Antigua and Barbuda is a country in the English-speaking Leeward Islands with a projected 2021 population of 99,337 [15], with a lone tertiary hospital that caters to most cancer cases diagnosed in the country [16]. Public healthcare is mostly financed through statutory deductions [17,18]. Colorectal cancer is ranked among the top five causes of cancer deaths locally [19], and its incidence reflects the general trends observed elsewhere in the Caribbean [20]. As such, there is a burgeoning need to understand this cancer’s economic impact.

As a country whose main contributor to GDP per capita is tourism [21], Antigua and Barbuda’s monetary policy and currency are guided by its relations with the Eastern Caribbean Central Bank and Eastern Caribbean Currency Union, respectively [22,23]. Moreover, for a country with a relatively high Human Capital Index [24], no study has ever been published on the extent of colorectal cancer’s economic burden there. Therefore, we aimed to calculate the economic burden of colorectal cancer in Antigua and Barbuda from the healthcare provider’s perspective by estimating its direct medical costs.

## 2. Materials and Methods

### 2.1. Study Design and Population

In this retrospective prevalence-based cost-of-illness study from the healthcare provider’s perspective, we used the data and/or results reported in the following publications: “Incidence, trends and patterns of female breast, cervical, colorectal and prostate cancers in Antigua and Barbuda, 2017–2021: a retrospective study” [25], “The economic burden of prostate cancer in Antigua and Barbuda: A prevalence-based cost-of-illness analysis from the healthcare provider perspective” [26], and “Cost analysis related to diagnosis, treatment and management of cervical cancer in Antigua and Barbuda: A prevalence-based cost-of-illness study” [16]. We used these data to estimate the direct medical costs of colorectal cancer in Antigua and Barbuda [16,26].

As stated elsewhere [25], we used data from patients diagnosed with colon and/or rectum cancer between 1 January 2017 and 31 December 2021, and who had their disease categorized based on the International Classification of Diseases, 10th version (ICD-10), codes C18, C19, and C20 [25,27]. Data were abstracted from patient records at the Sir Lester Bird Medical Centre (SLBMC), Antigua and Barbuda; the Cancer Centre Eastern Caribbean (TCCEC); and the Medical Benefits Scheme (MBS) [25]. We also received data on colorectal-cancer-related deaths from the Health Information Division (HID), Ministry of Health, Antigua and Barbuda [25]. To prevent any inaccuracies in our estimates, we refrained from collecting data on cases with recurrent disease [25].

### 2.2. An Overview of Colorectal Cancer Management in Antigua and Barbuda (2017–2021)

Oncology care in this country follows cancer control and prevention strategies proposed by the World Health Organization (WHO) [26,28]. Coupled with adaptations of the National Comprehensive Cancer Network Clinical Practice Guidelines (NCCN Guidelines) in oncology, these guidelines are used to determine the treatment of many locally occurring cancers [29,30]. The key benefits of using these guidelines in the local context have been stated elsewhere [31].

Given that there is no health policy or screening protocol for colorectal cancer in Antigua and Barbuda [32], colorectal cancer management usually begins with either pre-referral or referral based on age (opportunistic if the person is >50 years old), symptomology, and physical examination [33]. Diagnosis requires examination of the entire colon using endoscopy, colonoscopy, or rigid sigmoidoscopy and may involve taking a tissue sample for biopsy [33]. In many cases, initial investigations may include a fecal occult blood test (FOBT) [33]. If the biopsy comes back positive for malignant cells, then staging is performed to ascertain the extent of the disease and to decide on the appropriate care strategy or treatment plan [33].

Staging is centered on imaging studies (double-contrast barium enema and computed tomography scan), in addition to pathological examinations of the resected specimen [33]. Treatment is stage-dependent and primarily involves surgery for localized colorectal cancer, chemotherapy, radiation, or a combination of these (chemoradiation, adjuvant, or neoadjuvant radiation) (Figure 1) [33]. Patients presenting with more advanced disease may require surgical liver resection, in addition to systemic therapy, radiation therapy, and targeted treatment [29,30]. Bone scanning and positron emission tomography (PET scan) are not currently available in Antigua and Barbuda [26,34]. Patients with colorectal cancer in need of these services might be treated in healthcare facilities outside of the country with the support of the Medical Benefits Scheme [26].

The costs related to all aspects of colorectal cancer management based on 2021 market prices were considered for this study, as derived from the sources highlighted in Table 1 [16].

### 2.3. Costing and Cost Analysis

The cost components used in this study were determined based on the healthcare interventions required in managing colorectal cancer. This included components linked to the screening, diagnosis, treatment, and management of this disease [35]. These were identified using a micro-costing approach, quantified and valued per case, and extrapolated using prevalence data to estimate national costs [16,36]. We collected data retrospectively using patient charts and Excel spreadsheets designed based on the general approach used locally with respect to colorectal cancer screening, diagnosis, and treatment (Figure 1) [16]. Data on the costs related to managing colorectal cancer, including diagnosis and treatment, were sourced from reimbursement records filed with the Medical Benefits Scheme as claims for medical services, such as lab work, surgeries, and medication services, delivered by private healthcare entities in Antigua and Barbuda [18,26,37]. Supplementary cost information on our cost parameters was obtained from private health facilities in Antigua and Barbuda [26].

Premised on the usefulness of assessing disease burden [38,39] and the convenience of employing a top-down and bottom-up approach [10], our investigation of direct medical costs considered recurrent costs [36]. This included costs linked with personnel, travel, consumables, supplies (medical and non-medical), treatment, administration, and overheads [36].

Our cost analysis utilized an approach mentioned in previous studies [40]:Direct Medical Costs of Disease (dc) = ∑(Hi × Ri)
where
Hi is the number of cases requiring healthcare;Ri is the required healthcare resource unit costs per case;dc is the total cost.

The costs are reported in 2021 USD. We calculated them by accounting for the consumer price index (*CPI*) of 2021 and the 2021 USD exchange rate (1 USD = 2.7169 XCD):$$Value\;in\;2021\;USD=base\;year\;price\times \frac{CPI\;in\;2021}{CPI\;in\;based\;year}$$

*CPI* in 2021 = 95.27; *CPI* in base year = 95.27 [41].

The total costs were computed as follows: Direct Medical Costs of Disease (dc) = ∑(Hi × Ri)

Consistent with the approach stated elsewhere, we estimated the number of colorectal cancer cases requiring healthcare per year in Antigua and Barbuda by subtracting the number of deaths from the total number of cases and dividing the answer by 5 [36]. We determined the effect of the unrecorded cases at our study sites on our total cost estimates by increasing our average prevalent cases by 50%. We also assessed the effect of change on the total costs by reducing the treatment costs in the initial model by 50% [16]. Moreover, we analyzed the direct medical costs related to colon and rectal cancer cases when viewed separately. This also required increasing the average prevalent cases of colon cancer by 50% and those of rectal cancer by 300% to further assess the effect of unrecorded cases of these cancers on our estimates.

For comparison, our estimates of total costs were reported with and without services considered healthcare imports (outsourced services) [16]. These were PET scans and their associated transportation and accommodation costs [16,42].

### 2.4. Sensitivity Analysis

Applying an approach previously adopted and discussed in other studies, we conducted a sensitivity analysis to account for uncertainties in the derived cost estimates or possible inaccuracies in the data analysis [10,16,36,43]. Sensitivity analysis was also performed to account for any unrecorded cases at the study sites involved in our research. This was achieved by varying our cost estimates by a range of ±25% [10,16,36,43].

### 2.5. Ethical Considerations

This study was approved by the Antigua and Barbuda Institutional Review Board, Ministry of Health (AL-04/052022-ANUIRB); the Institutional Review Board of the Sir Lester Bird Medical Centre, Antigua; and the University of KwaZulu-Natal Biomedical Research Ethics Committee (BREC/00004531/2022). Patient and cost data were gathered between 16 September 2022 and 16 January 2023 and between 22 November 2022 and 25 January 2024, respectively. There was no direct contact with the cases studied, nor was there any risk posed to the patients [25]. The patient identities were preserved, as we did not record their names at any point during or after data collection [25].

## 3. Results

### 3.1. General Information

Table 2 shows that the mean (±SD) age at which a patient was seen by a specialist was 65.2 (±12.1) years [25]. Roughly 24% were diagnosed between the ages of 35 and 54, while approximately 51% were between the ages of 55 and 74, and 24% were ≥75 years [25].

In total, 42% of cases had early-stage disease (stages I and II), while the remaining 58% had late-stage disease (stages III and IV) [44,45].

Examined separately, there were 72 cases of colon cancer and 7 cases of rectal cancer diagnosed in the 2017–2021 period (Table 2).

### 3.2. Estimate of Colorectal Cancer Cases per Year

The information obtained from our study sites indicates that 22 of the 79 diagnosed cancer cases died during the study period [25], including 20 colon cancer cases and 2 rectal cancer cases. Therefore, we estimated that there were 11 cases of colorectal cancer on average per year in Antigua and Barbuda. We determined this by subtracting the 22 deaths from the 79 cases diagnosed in the period and dividing the result (57) by 5. This is further explained by the equation below:$$\mathrm{Patients}\;\mathrm{in}\;a\;\mathrm{single}\;\mathrm{year}\;(\mathrm{Cal})=\frac{\left(Cc-Da\right)}{5}$$
where

*Cc* is the number of diagnosed cases of colorectal cancer (2017–2021) = 79;*Da* is the number of patients with colorectal cancer who were diagnosed between 2017 and 2021 and died in the same period = 22;Cal is the average prevalent cases (patients in a single year) = 11.

Moreover, when the average number of diagnosed cases in a single year increased by 50%, the estimated cases per year became Cal = 17.

Consistent with the approach mentioned above, the average number of diagnosed colon cancer cases was estimated as 10 per year. For rectal cancer, this was initially computed to be one per year, increasing to four per year after adjusting this value by 300% for reasonable analysis and contrast.

### 3.3. Direct Medical Unit Costs

Table 3 shows the estimates of the direct medical unit costs for managing colorectal cancer in 2021. Overall, this cost was USD 139,295.58 (Table 3A), with the leading three drivers being treatment (USD 83,567.96), post-treatment side-effects care (USD 39,646.09), and other direct costs (USD 11,439.68) (Table 3A) (Figure 2). These costs fluctuated based on clinical stages, starting at USD 67,631.06 for stage I and increasing to USD 94,742.65, USD 111,480.61, and USD 99,792.15 for stages II, III, and IV, respectively (Table 3A,B) (Figure 3).

The main drivers of the overall direct medical unit costs were surgery (USD 43,467.10), other complications of treatment (USD 28,469.72), and immunotherapy (USD 19,200.00) (Table 3).

In comparison, the overall direct medical unit costs of colon and rectal cancers were USD 125,796.86 and USD 121,723.50, respectively, with marginally higher unit costs for each stage of rectal cancer compared with colon cancer (Table 3A).

### 3.4. Total Annual Direct Medical Costs

The total annual direct medical costs for colorectal cancer were estimated to be USD 1,138,674.08 (ranging between USD 854,005.56 and USD 1,423,342.60). Treatment accounted for the largest share of annual direct medical costs at 54% (USD 613,650.01; ranging from USD 460,237.51 to USD 767,062.51) (Table 4) (Figure 4). The second major contributor to annual direct medical costs was post-treatment side-effects care at 35% (USD 402,234.50; ranging from USD 301,675.88 to USD 502,793.13) (Table 4) (Figure 4). The next highest contributors to annual direct medical costs were diagnosis, imaging, and other direct costs at roughly 4% (USD 49,281.21; ranging from USD 36,960.91 to USD 61,601.51 and USD 42,130.42; ranging from USD 31,597.82 to USD 52,663.03, respectively). The remaining care parameter accounted for 3% of the annual direct medical costs (Table 4).

Following an increase in the average number of prevalent cases by 50% (from 11 to 17 for a single year), the estimated total annual direct medical costs became USD 1,741,985.51 (ranging between USD 1,306,489.13 and USD 2,177,481.89), representing a 53% increase in the total annual direct medical costs (Table 5). With this increase in prevalence, we found that treatment, post-treatment side-effects care, and diagnosis and imaging remained the major drivers of annual direct medical costs (Table 5) (Figure 5).

Viewed separately and for the purposes of comparison, the total annual direct medical costs for colon and rectal cancers were estimated to be USD 1,010,503.44 (ranging between USD 757,877.58 and USD 1,263,129.30) and USD 394,816.64 (ranging between USD 296,112.48 and USD 493,520.80), respectively (Table 4). When the average prevalent cases were increased by 50%, the estimates became USD 1,498,618.58 (ranging between USD 1,123,963.94 and USD 1,873,273.23) and USD 594,789.47 (ranging between USD 446,092.10 and USD 743,486.84) for colon and rectal cancer, respectively (Table 5).

## 4. Discussion

This study provides up-to-date evidence regarding the economic burden of colorectal cancer in Antigua and Barbuda. The findings indicate that the estimated annual direct medical costs and economic burden of colorectal cancer in 2021 were USD 1.14 million, with the major drivers of costs being treatment (USD 613,650.01) and post-treatment side-effects care (USD 402,234.50). Given that this is a population where the average life expectancy at birth in 2021 was 78 [46], and the median age at diagnosis for colorectal cancer was 67, our findings suggest that the impact of an aging population could be a contributing factor to the disease’s prevalence and, ultimately, its observed cost burden [47]. The identified annual direct medical costs are considerable when assessed in the context of the country’s health expenditure per capita in 2021, which was USD 923.41 [48] and, in the context of its budgeted national health allocations for 2021, approximately 1% [49]. The results indicate that the high annual direct medical costs of treatment, although commensurate with the disease stage, hinge on the completion of surgery as a starting point, followed by other care components (chemotherapy, radiation therapy, and/or immunotherapy) as part of the general armamentarium used to address issues related to cures and/or disease progression [47,50]. Additionally, the high annual direct medical costs of post-treatment side-effects care could be attributed to the direct effects of surgery and post-surgical care, among other things. This could be linked with having to manage dangerous conditions, such as anastomotic leakage, as well as other such postoperative complications associated with the inflammatory response to colorectal surgery [51].

The cost of diagnosing, detecting, and treating colorectal cancer varies significantly across countries and regions of the world [11,50]. This is due to large variations in the methodologies, types of estimated costs, and models used to compare the results of cost-of-illness studies [12,50]. Notwithstanding this difficulty, our results suggest that the economic burden of colorectal cancer may be commensurate with the size of the local population and particular to the local context, which is consistent with observations made in several studies. For instance, the results suggest that stage I disease had the lowest direct medical unit costs, while stage III disease was the costliest. This observation was consistent with previous studies conducted in Ireland by Tilson et al. and in Iran by Davari et al. [52,53]. A similar observation was also noted for annual direct medical costs, which were directly linked with both direct medical unit costs and corresponded to the number of cases per disease stage. Although costs tend to increase with disease progression, this observation could have arisen due to treatment delays and/or other delays, which could cause persons with stage III disease having a greater chance of receiving more cancer care-related services, with more inherent difficulties in the prognosis, treatment, and costs of associated services than persons with the other disease stages [53].

Alongside our analysis of colorectal cancer estimates, our examination of component cancers in colon and rectal cancers suggests that variations in the patterns of each cancer are related to differences in their estimates of direct medical costs [54]. Future studies could investigate this observation further.

There is a lack of knowledge about the economic burden of colorectal cancer in Antigua and Barbuda. With the projected rise in the incidence and prevalence of this cancer in years to come [55], its economic burden can be expected to increase significantly [12]. Therefore, our results imply that introducing cost-containment initiatives could help reduce the economic burden of this disease in the coming years. Initiatives could focus on (i) engendering greater health-seeking behavior through various disease-sensitization campaigns that, among other things, encourage people to adopt healthy lifestyle behaviors, including adjustments to diet, engaging in regular physical activity, and seeking timely colorectal cancer screening [56]. These initiatives could also focus on (ii) encouraging clinicians to keep abreast of improvements in colorectal cancer management practices through regular participation in tumor board meetings and other medical education fora [57]. Initiatives could also be directed at reviewing the current approach to colorectal cancer care locally, in order to optimize the quality of care and thereby reduce the economic costs of managing this disease [56]. Furthermore, as cost studies have shown that treatment costs in the early stages of this disease are much lower than those in later stages and because stage III disease was costlier in our study, our findings could serve as a basis for advocating for both the establishment of a national colorectal cancer screening program and making improvements in our healthcare infrastructure as ways of diagnosing this disease at the early stages. This may warrant instituting better drug selection and procurement practices, including replacing expensive chemotherapeutic agents with equally therapeutic or more efficacious low-cost alternatives; improving facilities for screening, early detection, diagnosis and treatment; and improving the capacity of facilities by hiring or training a cadre of health professionals to cater to more affordable surgeries [47]. Given the general distribution of cases by age groups, besides being useful as baseline information for improving existing preventative and screening plans to mitigate the threat of colorectal cancer in our elderly population in the future [12], our results could be useful in determining which care strategies or approaches are cost-effective [47]. This could shift the current stage of distribution for this cancer to a more acceptable pattern [47]. Moreover, besides providing information that can guide the development of health policies, our study may serve as a basis for (i) determining future research priorities regarding colorectal cancer, (ii) assessing new treatment modalities in order to review drug treatment protocols, and (iii) advocating for affordable surgical care, among other things [53].

This study has several notable strengths. First, to the best of our knowledge, this is the first study of colorectal cancer costs in Antigua and Barbuda. It presents a comprehensive assessment and analysis of cost data and the costs of care components that impact the economic burden of colorectal cancer in this country. This information is very detailed and reflects the actual direct costs of diagnosing, treating, and managing colorectal cancer from the healthcare provider’s perspective. Similar to observations made by Bovell and colleagues [16,25], another strength of this study is its relevance in highlighting the lack of a national (i) cancer registry, (ii) register of costs, (iii) colorectal cancer screening program, and (iv) legislation to aid in compliance with reporting colorectal cancer cases at all appropriate levels of healthcare on the island [26]. A notable strength is that our study was aided by information from expert health practitioners working locally in treating and managing colorectal cancer. This ensured that the study did not lack technical and/or other background information, the presence of which helped validate our study findings.

Concerning limitations, because this study utilized the prevalence-based method of the cost-of-illness approach, with a focus on direct medical costs, our list of care components under parameters such as post-treatment side-effects care, ongoing care, and other direct costs was not entirely exhaustive. This means that cost elements, such as end-of-life care, productivity losses, and other indirect medical costs, were not considered in our cost estimates [58]. Along with possible inherent data limitations owing to record abstraction, this could have led to an underestimation of the economic burden of colorectal cancer. Future studies could incorporate these elements to provide a more robust view of the economic burden of this condition. Notwithstanding these limitations, however, we are confident that our study’s findings—while also useful for comparisons in other similar settings and with estimates of other solid organ tumors [59]—reflect the current situation regarding the cost of illness for colorectal cancer in Antigua and Barbuda.

## 5. Conclusions

Our study presents up-to-date evidence regarding the economic burden of colorectal cancer in Antigua and Barbuda. The estimates of annual direct medical costs appear to be substantial given the local context, with the major drivers being treatment (54%) and post-treatment side-effects care (35%). Employing cost containment measures, such as revising drug procurement practices, could immensely reduce these costs going forward. Optimizing data collection via a national cancer registry and register of costs, as well as engaging in studies using cost-effectiveness and cost-utility models, could allow for more extensive cost estimates in the future.

Moreover, notwithstanding the usefulness of our cost estimates to health administrators for (i) designing new and/or improving existing policies, (ii) assisting in budget planning, and (iii) implementing guidelines for allocating resources used in colorectal cancer care locally, these health management parameters could be better informed should future cost-of-illness studies consider productivity losses and other elements of indirect costs in assessing the economic burden of colorectal cancer in Antigua and Barbuda.

## Figures and Tables

**Figure 1 ijerph-22-00552-f001:**
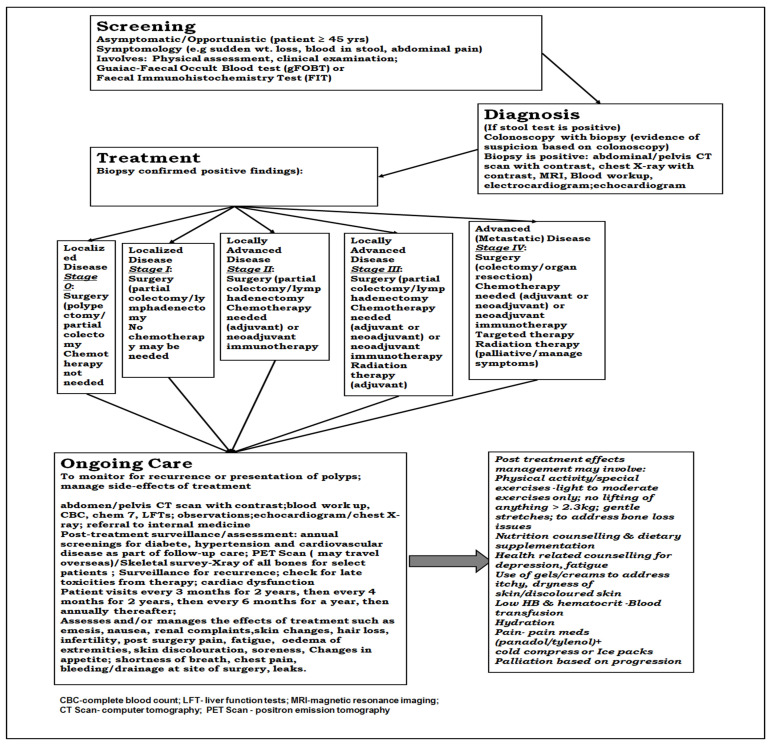
Schematic illustration of the care components in the management of colorectal cancer in Antigua and Barbuda.

**Figure 2 ijerph-22-00552-f002:**
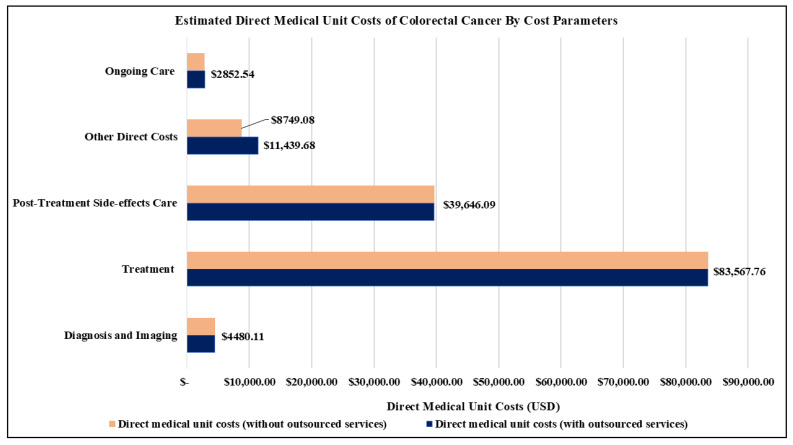
Key contributors to direct medical unit costs of colorectal cancer.

**Figure 3 ijerph-22-00552-f003:**
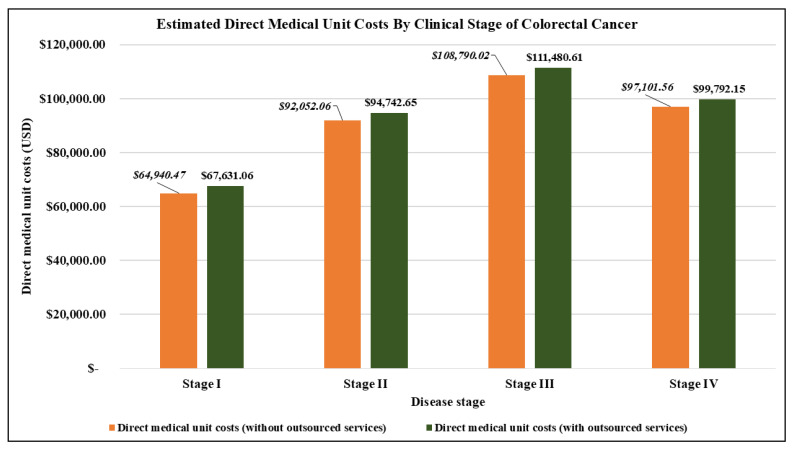
Direct medical unit costs categorized by disease stage (stages I–IV).

**Figure 4 ijerph-22-00552-f004:**
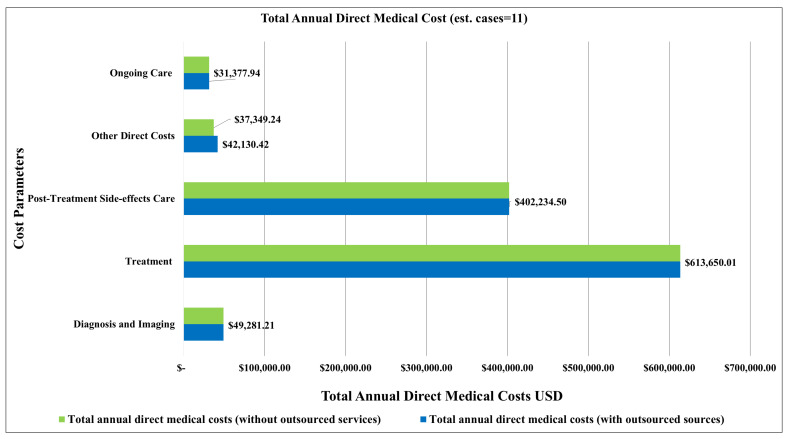
Total annual direct medical costs of colorectal cancer broken down by care parameters and estimates of cases in a single year (n = 11).

**Figure 5 ijerph-22-00552-f005:**
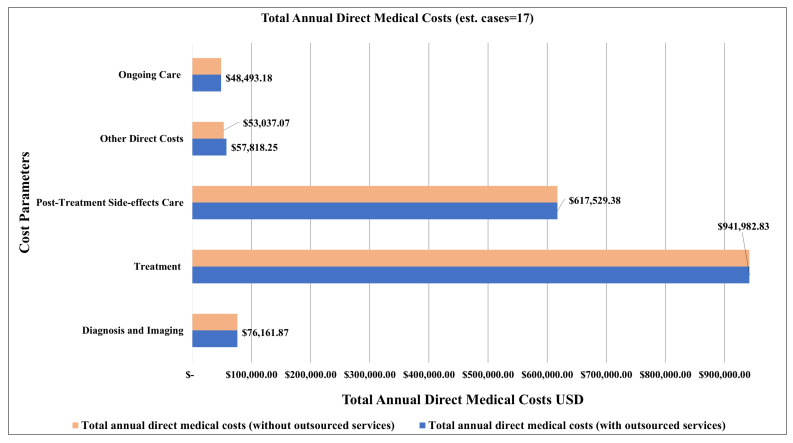
Total annual direct medical costs of colorectal cancer broken down by care parameters and estimates of cases in a single year (n = 17).

**Table 1 ijerph-22-00552-t001:** Data categories, variables, and sources of costs.

**Data/Parameter**	**Data Source**	**Price Source**
Estimated number of colorectal cancer cases	Aggregate data on cancer cases were extracted from patient records of The Cancer Centre of the Eastern Caribbean, the Sir Lester Bird Medical Centre, and the Medical Benefits Scheme (2017–2021), alongside death-related data from the Ministry of Health	N/A
Screening		
Consultation	Consultation with specialist physicians at The Cancer Centre Eastern Caribbean, the Medical Benefits Scheme, and the Sir Lester Bird Medical Centre. Reviewed claims and refund records at the Medical Benefits Scheme.	Market price
Clinical assessment		Market price
Physical assessment		Market price
Guaiac fecal occult blood test (gFOBT)	Consultation with private laboratories on service fees charged. Reviewed claims and refunds records at the Medical Benefits Scheme.	Market price/private laboratory
Diagnosis		
Colonoscopy	Consultation with specialist physicians at The Cancer Centre Eastern Caribbean, the Medical Benefits Scheme, and the Sir Lester Bird Medical Centre. Reviewed claims and refund records at the Medical Benefits Scheme. Consultation with finance experts at the Medical Benefits Scheme and the Sir Lester Bird Medical Centre. Consultation with private health facilities.	Market price
Endoscopy		
Complete blood workup		
Histopathology		Market price/private laboratory
Imaging studies (computed tomography (CT scan); X-ray)		Market price
EKG/ECG		Market price/private clinic
Treatment		
Surgery (colectomy)	Consultation with specialist physicians at The Cancer Centre Eastern Caribbean, the Sir Lester Bird Medical Centre, and private health facilities and pharmacies. Reviewed claims and refund records at the Medical Benefits Scheme. Consultation with finance experts at the Medical Benefits Scheme, The Cancer Centre Eastern Caribbean, and the Sir Lester Bird Medical Centre.	
Surgery (lymphadenectomy)		Market price
Surgery (organ resection—liver/lungs)		Market price
Chemotherapy (neoadjuvant/adjuvant)		Market price
Immunotherapy		Market price/ private supplier
Radiation therapy		Market price
Post-treatment effects care		
Urinary retention	Consultation with specialist physicians at The Cancer Centre Eastern Caribbean, the Sir Lester Bird Medical Centre and private health facilities and pharmacies. Reviewed claims and refund records at the Medical Benefits Scheme. Consultation with finance experts at the Medical Benefits Scheme, The Cancer Centre Eastern Caribbean, and the Sir Lester Bird Medical Centre.	Market price
Urinary incontinence		Market price
Renal complaints		Market price
Lower urinary tract infections		Market price
Other complications of treatment		Market price
Other direct costs		
Nutrition counselling	Consultation with specialist physicians at The Cancer Centre Eastern Caribbean, the Sir Lester Bird Medical Centre, and private health facilities and pharmacies. Reviewed claims and refund records at the Medical Benefits Scheme. Consultation with finance experts at the Medical Benefits Scheme, The Cancer Centre Eastern Caribbean, and the Sir Lester Bird Medical Centre.	Private/ market price
Psychiatric/psychological counselling		Market price
Pharmacy services		Private pharmacy/market price
Imaging (PET scan—overseas)		MBS-approved financing/market price
Emergency kit		
Transportation/accommodation (imaging/treatment—overseas)		MBS-approved financing/market price
Transportation (local services related)		Market price
Overheads		Market price

N/A—not applicable.

**Table 2 ijerph-22-00552-t002:** Background attributes of the population of cases of colorectal cancer (combined and disaggregated) (2017–2021).

Attributes	Colorectal Cancer N = 79, N (%)	Colon Cancer N = 72 N (%)	Rectal Cancer N = 7, N (%)
Age at presentation			
Mean age (SD)	65.2 (12.1)	65.7 (11.9)	60.3 (14.3)
Mean age 95%CI	62.5–67.9	62.9–68.5	47.1–73.5
Median age (IQR)	67.0 (20.0)	67.0 (21.0)	65.0 (19.0)
Age range	32.0–87.0	42.0–87.0	32.0–73.0
Age distribution			
20–34	1 (1.3)	-	1 (14.3)
35–54	19 (24.1)	18 (25.0)	1 (14.3)
55–74	40 (50.6)	35 (48.6)	5 (71.4)
≥75	19 (24.1)	19 (26.4)	-
Clinical stage			
I	15 (19.0)	13 (18.1)	2 (28.6)
II	18 (22.8)	16 (22.2)	2 (28.6)
III	33 (41.8)	32 (44.4)	1 (14.3)
IV	13 (16.5)	11 (15.3)	2 (28.6)

**Table 3 ijerph-22-00552-t003:** (**A**) Costs for staging, management, and treatment of colorectal cancer stages I–IV. (**B**) Range of costs for staging, management, and treatment of colorectal cancer stages I–IV.

(A)						
Staging and Treatment Variables		Unit Costs (USD)	Clinical Stage			
Estimated Number of Cases in a Single Year (N = 11)			I (n = 2)	II (n = 3)	III (n = 5)	IV (n = 1)
Diagnosis and imaging						
Consultation (clinical assessment, physical examination)		147.23	147.23	147.23	147.23	147.23
Guaiac fecal occult blood test		14.72	14.72	-	-	-
Colonoscopy		1288.23	1288.23	1288.23	1288.23	1288.23
Biopsy		368.07	368.07	368.07	368.07	368.07
Imaging (radiology)		1503.18	1503.18	1503.18	1503.18	1503.18
Laboratory		530.02	530.02	530.02	530.02	530.02
Histopathology		628.66	628.66	628.66	628.66	628.66
Treatment (average colorectal cancer cases in a single year = 11)						
Surgery (colectomy)		20,262.68	20,262.68	20,262.68	20,262.68	-
Surgery (lymphadenectomy)		7315.10	7315.10	7315.10	7315.10	-
Surgery (resection)		15,889.32	-	-	-	15,889.32
Radiotherapy (external beam radiotherapy (EBRT))		12,974.35	-	-	12,974.35	12,974.35
Systemic therapy (chemotherapy) *		7926.31	-	7926.31	7926.31	7926.31
Immunotherapy		19,200.00	-	19,200.00	19,200.00	19,200.00
Post-treatment side-effects care						
Blood clot prophylaxis		360.00	360.00	360.00	360.00	360.00
Renal complaints		3763.61	-	-	3763.61	3763.61
Anemia (low hemoglobin/hematocrit)		6687.76	6687.76	6687.76	6687.76	6687.76
Infection control		365.00	290.42	290.42	290.42	290.42
Other complications of treatment (post-operative pain, wound care, other issues, etc.)		28,469.72	13,942.77	13,942.77	13,942.77	13,942.77
Other direct costs						
Nutrition counselling		100.00	100.00	100.00	100.00	100.00
Psychiatric/psychological counselling		128.82	128.82	128.82	128.82	128.82
Pharmacy services		89.99	89.99	89.99	89.99	89.99
Positron emission tomography (PET) scan (overseas) **		991.94	991.94	991.94	991.94	991.94
Chemotherapy port insertion		7361.33	7361.33	7361.33	7361.33	7361.33
Emergency kit (chemo)		470.83	470.83	470.83	470.83	470.83
Patient transportation/accommodation (overseas imaging) **		1398.65	1398.65	1398.65	1398.65	1398.65
Transportation (local)		561.30	561.30	561.30	561.30	561.30
Overheads		36.81	36.81	36.81	36.81	36.81
Ongoing care						
Follow-up consultations		368.07	368.07	368.07	368.07	368.07
Imaging studies (CT scan, chest X-ray, echocardiogram)		975.38	975.38	975.38	975.38	975.38
Biochemistry tests (chemistry/renal panel, liver function tests, HbA1c, cholesterol)		1509.09	1509.09	1509.09	1509.09	1509.09
Total (crude estimates)		139,295.58	64,940.47	92,052.06	108,790.02	**97,101.56**
Total (revised estimates)		141,986.18	67,631.06	94,742.65	111,480.61	**99,792.15**
Total (revised estimates)-C		125,796.86	67,331.06	94,442.65	111,180.61	**83,602.83**
Total (revised estimates)-R		121,723.50	76,232.05	103,343.64	107,107.25	**83,902.83**
(**B**)						
**Clinical Stage**	**Care Parameters**	**Direct Medical** **Unit Costs (USD)**	**Range (USD) ±25%**			
			**Lower**		**Upper**	
1	Diagnosis and imaging	4480.11	3360.08		5600.14	
	Treatment	27,577.78	20,683.34		34,472.23	
	Post-treatment side-effects care	21,280.95	15,960.71		26,601.19	
	Other direct costs	11,439.68	3058.76		5097.93	
	Ongoing care	2852.54	2139.41		3565.68	
	Total	67,631.06	50,723.30		84,538.83	
2	Diagnosis and imaging	4465.39	3349.04		5581.74	
	Treatment	54,704.09	41,028.07		68,380.11	
	Post-treatment side-effects care	21,280.95	15,960.71		26,601.19	
	Other direct costs	11,439.68	3058.76		5097.93	
	Ongoing care	2852.54	2139.41		3565.68	
	Total	94,742.65	71,056.99		118,428.31	
3	Diagnosis and imaging	4465.39	3349.04		5581.74	
	Treatment	67,678.44	50,758.83		84,598.05	
	Post-treatment side-effects care	25,044.56	18,783.42		31,305.70	
	Other direct costs	11,439.68	3058.76		5097.93	
	Ongoing care	2852.54	2139.41		3565.68	
	Total	111,480.61	83,610.46		139,350.76	
4	Diagnosis and imaging	4465.39	3349.04		5581.74	
	Treatment	55,989.98	41,992.49		69,987.48	
	Post-treatment side-effects care	25,044.56	18,783.42		31,305.70	
	Other direct costs	11,439.68	3058.76		5097.93	
	Ongoing care	2852.54	2139.41		3565.68	
	Total	99,792.15	74,844.11		124,740.19	
Overall direct medical unit cost (revised estimates)	Diagnosis and imaging	4480.11	3360.08		5600.14	
	Treatment	83,567.76	62,675.82		104,459.70	
	Post-treatment side-effects care	39,646.09	29,734.57		49,557.61	
	Other direct costs	11,439.68	11,439.68		11,439.68	
	Ongoing care	2852.54	2139.41		3565.68	
	Total	141,986.18	106,489.64		177,482.73	

* based on medicines used in actual treatment regimens. ** Outsourced cancer care services linked to overseas care. Crude estimates: The costs of outsourced cancer care services were excluded from the analysis. Revised estimates: The costs of outsourced cancer care services were included in the analysis. Revised estimates—C: Computed direct medical unit costs for colon cancer-specific care, with the costs of outsourced cancer care services included in the analysis (see Appendix A). Revised estimates—R: Computed direct medical unit costs for rectal cancer-specific care, with the costs of outsourced cancer care services included in the analysis (see Appendix A).

**Table 4 ijerph-22-00552-t004:** Total annual cost estimation for colorectal cancer (direct medical costs) (estimated cases = 11).

Parameter	Care Component/Procedures	Average Number of Cases in a Single Year (N = 11)	Estimated Average Cost 2021 (USD)	Total Costs (USD)	Sum Total and Percentage of Cost (Adjusted)	Range (USD) ± 25%	
						Lower	Upper
Diagnosis and imaging	Diagnosis and imaging						
	Consultation (clinical assessment/physical examination)	11	147.23	1619.53		1214.65	2024.41
	Guaiac fecal occult blood test	11	14.72	161.92		121.44	202.40
	Colonoscopy	11	1288.23	14,170.53		10,627.90	17,713.16
	Biopsy	11	368.07	4048.77		3036.58	5060.96
	Imaging (radiology)	11	1503.18	16,534.98		12,401.24	20,668.73
	Laboratory	11	530.02	5830.22		4372.67	7287.78
	Histopathology	11	628.66	6915.26		5186.45	8644.08
Subtotal				49,281.21	4.33%	36,960.91	61,601.51
Treatment	Treatment						
	Stage I	2	27,577.78	55,155.56		41,366.67	68,944.45
	Stage II	3	54,704.09	164,112.27		123,084.20	205,140.34
	Stage III	5	67,678.44	338,392.20		253,794.15	422,990.25
	Stage IV	1	55,989.98	55,989.98		41,992.49	69,987.48
Subtotal				613,650.01	53.89%	460,237.51	767,062.51
Post-treatment care	Post-treatment care						
	Blood clot prophylaxis	11	360.00	3960.00		2970.00	4950.00
	Renal complaint	2	3763.61	7527.22		5645.42	9409.03
	Anemia (low hemoglobin/hematocrit)	11	6687.76	73,565.36		55,174.02	91,956.70
	Infection control	11	365.00	4015.00		3011.25	5018.75
	Other complications of treatment	11	28,469.72	313,166.92		234,875.19	391,458.65
Subtotal				402,234.50	35.32%	301,675.88	502,793.13
Other direct medical costs	Other direct costs						
	Nutrition counselling	11	100.00	1100.00		825.00	1375.00
	Psychiatric/psychological counselling	11	128.82	1417.02		1062.77	1771.28
	Pharmacy services	11	89.99	989.89		742.42	1237.36
	Positron emission tomography (PET) Scan (overseas) **	2	991.94	1983.88		1487.91	2479.85
	Chemotherapy port insertion	3	7361.33	22,083.99		16,562.99	27,604.99
	Emergency kit (chemo)	11	470.83	5179.13		3884.35	6473.91
	Patient Transportation/accommodation (overseas imaging) **	2	1398.65	2797.30		2097.98	3496.63
	Transportation (local)	11	561.30	6174.30		4630.73	7717.88
	Overheads	11	36.81	404.91		303.68	506.14
Subtotal				42,130.42	3.70%	31,597.82	52,663.03
Ongoing care	Ongoing care						
	Follow-up consultations	11	368.07	4048.77		3036.58	5060.96
	Imaging studies (CT scan, chest X-ray, echocardiogram)	11	975.38	10,729.18		8046.89	13,411.48
	Biochemistry tests (chemistry/renal panel, liver function tests, HbA1c, cholesterol)	11	1509.09	16,599.99		12,449.99	20,749.99
Subtotal				31,377.94	2.76%	23,533.46	39,222.43
Total direct medical costs (crude estimates)				1,133,892.90		850,419.68	1,417,366.13
Total direct medical costs (revised estimates)				1,138,674.08		854,005.56	1,423,342.60
Total direct medical costs (revised estimates)—C				1,010,503.44		757,877.58	1,263,129.30
Total direct medical costs (revised estimates)—R				394,816.64		296,112.48	493,520.80

** Outsourced cancer care services linked to overseas care. Crude estimates: The costs of outsourced cancer care services were excluded from the analysis. Revised estimates: The costs of outsourced cancer care services were included in the analysis. Revised estimates—C: Computed total annual direct medical unit costs for colon cancer-specific care, with the costs of outsourced cancer care services included in the analysis and assuming an average prevalence of 10 cases per year (see Appendix A). Revised estimates—R: Computed total annual direct medical unit costs for rectal cancer-specific care, with the costs of outsourced cancer care services included in the analysis and assuming an average prevalence of 4 cases per year (see Appendix A).

**Table 5 ijerph-22-00552-t005:** Total annual cost estimation for colorectal cancer (direct medical costs) (estimated cases = 17) (50% increase in average prevalence).

Parameter	Care Component/Procedures	Average Number of Cases in a Single Year (N = 17)	Estimated Average Cost 2021 (USD)	Total Costs (USD)	Sum Total and Percentage of Cost (Adjusted)	Range (USD)± 25%	
						Lower	Upper
Diagnosis and imaging	Diagnosis and imaging						
	Consultation (clinical assessment/physical examination)	17	147.23	2502.91		1877.18	3128.64
	Guaiac fecal occult blood test	17	14.72	250.24		187.68	312.80
	Colonoscopy	17	1288.23	21,899.91		16,424.93	27,374.89
	Biopsy	17	368.07	6257.19		4692.89	7821.49
	Imaging (radiology)	17	1503.18	25,554.06		19,165.55	31,942.58
	Laboratory	17	530.02	9010.34		6757.76	11,262.93
	Histopathology	17	628.66	10,687.22		8015.42	13,359.03
Subtotal				76,161.87	4.37%	57,121.40	95,202.34
Treatment	Treatment						
	Stage I	3	27,577.78	82,733.34		62,050.01	103,416.68
	Stage II	5	54,704.09	273,520.45		205,140.34	341,900.56
	Stage III	7	67,678.44	473,749.08		355,311.81	592,186.35
	Stage IV	2	55,989.98	111,979.96		83,984.97	139,974.95
Subtotal				941,982.83	54.08%	706,487.12	1,177,478.54
Post-treatment care	Post-treatment care						
	Blood clot prophylaxis	17	360.00	6120.00		4590.00	7650.00
	Renal complaint	2	3763.61	7527.22		5645.42	9409.03
	Anemia (low hemoglobin/hematocrit)	17	6687.76	113,691.92		85,268.94	142,114.90
	Infection control	17	365.00	6205.00		4653.75	7756.25
	Other complications of treatment	17	28,469.72	483,985.24		362,988.93	604,981.55
Subtotal				617,529.38	35.45%	463,147.04	771,911.73
Other direct medical costs	Other direct costs						
	Nutrition counselling	17	100.00	1700.00		1275.00	2125.00
	Psychiatric/psychological counselling	17	128.82	2189.94		1642.46	2737.43
	Pharmacy services	17	89.99	1529.83		1147.37	1912.29
	Positron emission tomography (PET) scan (overseas) **	2	991.94	1983.88		1487.91	2479.85
	Chemotherapy port insertion	4	7361.33	29,445.32		22,083.99	36,806.65
	Emergency kit (chemo)	17	470.83	8004.11		6003.08	10,005.14
	Patient transportation/accommodation (overseas imaging) **	2	1398.65	2797.30		2097.98	3496.63
	Transportation (local)	17	561.30	9542.10		7156.58	11,927.63
	Overheads	17	36.81	625.77		469.33	782.21
Subtotal				57,818.25	3.32%	43,363.69	72,272.81
Ongoing care	Ongoing care						
	Follow-up consultations	17	368.07	6257.19		4692.89	7821.49
	Imaging studies (CT scan, chest X-ray, echocardiogram)	17	975.38	16,581.46		12,436.10	20,726.83
	Biochemistry tests (chemistry/renal panel, liver function tests, HbA1c, cholesterol)	17	1509.09	25,654.53		19,240.90	32,068.16
Subtotal				48,493.18	2.78%	36,369.89	60,616.48
Total direct medical costs (crude estimates)				1,737,204.33		1,302,903.25	2,171,505.41
Total direct medical costs (revised estimates)				1,741,985.51		1,306,489.13	2,177,481.89
Total direct medical costs (revised estimates)—C				1,498,618.58		1,123,963.94	1,873,273.23
Total direct medical costs (revised estimates)—R				594,789.47		446,092.10	743,486.84

** Outsourced cancer care services linked to overseas care. Crude estimates: The costs of outsourced cancer care services were excluded from the analysis. Revised estimates: The costs of outsourced cancer care services were included in the analysis. Revised estimates—C: Computed total annual direct medical unit costs for colon cancer-specific care, with the costs of outsourced cancer care services included in the analysis and assuming an average prevalence of 15 cases per year (see Appendix A). Revised estimates—R: Computed total annual direct medical unit costs for rectal cancer-specific care, with the costs of outsourced cancer care services included in the analysis and assuming an average prevalence of 6 cases per year (see Appendix A).

## Data Availability

All data generated or analyzed during this study are included in the article. The data are fully available without restrictions, and inquiries can be directed to the corresponding author.

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
