# Peer review of "Analysis of the Direct Medical Costs of Colorectal Cancer in Antigua and Barbuda: A Prevalence-Based Cost-of-Illness Study"

_ijerph, 2025, doi:10.3390/ijerph22040552_

Round 1
Reviewer 1 Report
Comments and Suggestions for Authors
This retrospective, prevalence-based study attempts to identify the cost of illness associated with colorectal cancer in Antigua and Barbuda. The authors utilized methods they had utilized in previous publications to identify the direct medical costs associated with colorectal cancer management and in order to estimate the number of colorectal cancer cases they subtracted the number of deaths from the total number of cases and divided the outcome by 5. In addition, they performed a sensitivity analysis using the lower and upper bound of 25% to account for the impact of uncertainty in the cost estimation.
Their results showed that treatment, post-treatment side effects, and diagnosis/imaging account for the majority of medical costs and that stage III disease was associated with higher direct medical unit costs. Additionally, it is shown that the overall direct medical costs are driven by surgery, other complication, and immunotherapy.
Major Comment:
(1) In this study, colon and rectal cancer cases are "lumped" together. These are two different disease entities whose staging and management differ and therefore may be potentially associated with vastly different medical costs. This "lumping" of the two pathologies to a single disease entity may lead to under- or over- appreciation of the costs associated managing each disease. The authors should strongly consider performing two analyses, one for colon cancer and one for rectal cancer.
Minor comments:
(1) In the "Introduction" it appears as if colorectal cancer is a disease afflicting mainly individuals>50 years of age. However, early age of onset colorectal cancer is on the rise and constitutes a current healthcare concern. Therefore, the authors should consider adding some wording about the rising incidence of early age of onset colorectal cancer.
(2) The authors may want to consider moving the sentence "Staging on...specimen [24]" found in lines 93 to 95 after the sentence "In many cases... (FOBT) [24]" as this may potentially improve the flow of the manuscript.
(3) Please consider adding a breakdown of colon and rectal cancer cases in Table 1.
Author Response
Response to Reviewer 1 Comments
|
|||||||||||||||||||||||||||||||||||||||||||||||||||||||||||||||||||||
1. Summary |
|
|
|||||||||||||||||||||||||||||||||||||||||||||||||||||||||||||||||||
Thank you very much for taking the time to review this manuscript. We do express our appreciations to you for your comments and suggestions offered. It is our hope that the revised manuscript has addressed your concerns. We do look forward to hearing from you on this. Please find the detailed responses below and the corresponding revisions/corrections highlighted in track changes in the re-submitted files.
|
|||||||||||||||||||||||||||||||||||||||||||||||||||||||||||||||||||||
2. Point-by-point response to Comments and Suggestions for Authors
|
|||||||||||||||||||||||||||||||||||||||||||||||||||||||||||||||||||||
This retrospective, prevalence-based study attempts to identify the cost of illness associated with colorectal cancer in Antigua and Barbuda. The authors utilized methods they had utilized in previous publications to identify the direct medical costs associated with colorectal cancer management and in order to estimate the number of colorectal cancer cases they subtracted the number of deaths from the total number of cases and divided the outcome by 5. In addition, they performed a sensitivity analysis using the lower and upper bound of 25% to account for the impact of uncertainty in the cost estimation. Their results showed that treatment, post-treatment side effects, and diagnosis/imaging account for the majority of medical costs and that stage III disease was associated with higher direct medical unit costs. Additionally, it is shown that the overall direct medical costs are driven by surgery, other complication, and immunotherapy. Major Comment: Comment (1) In this study, colon and rectal cancer cases are "lumped" together. These are two different disease entities whose staging and management differ and therefore may be potentially associated with vastly different medical costs. This "lumping" of the two pathologies to a single disease entity may lead to under- or over- appreciation of the costs associated managing each disease. The authors should strongly consider performing two analyses, one for colon cancer and one for rectal cancer. Response (1): The authors wish to thank the reviewer for sharing this comment with us. After much contemplations and deliberations, the authors have decided to retain the table depicting the costs analyses of the colorectal cancer cases. This was decided upon since our computed estimates of rectal cancers cases per single year in the study period was less than 2. However, and so as not to lose the significance and/or importance of the reviewer’s suggestion, and while also showing contrast, we decided to improve the analysis by also accounting for any unrecorded colon cancer-specific and rectal cancer-specific cases by increasing the average prevalent cases of colon cancer by 50% and that of rectal cancer by 300% and thereafter reporting on the computed cost estimates. This information is highlighted in our text and tables. We also presented these estimates in supplementary tables accompanying the manuscript. In the end we not only presented computed estimates for both cancers combined but also for them as separate entities. See lines 155-158 Moreover, we analyzed the direct medical costs related to colon and rectal cancer cases when viewed separately. This also required increasing the average prevalent cases of colon cancer by 50% and rectal cancer by 300% to further assess the effect of unrecorded cases of these cancers on our estimates. And see inserts made in tables 3A, 4 and 5 respectively. See also supplementary files 1 and 2 for further reference. Minor comments: Comment (2) In the "Introduction" it appears as if colorectal cancer is a disease afflicting mainly individuals>50 years of age. However, early age of onset colorectal cancer is on the rise and constitutes a current healthcare concern. Therefore, the authors should consider adding some wording about the rising incidence of early age of onset colorectal cancer. Response (2): The authors have taken note of the reviewer’s comments and have since edited the area of the introduction section that addresses ‘colorectal cancer occurring in persons > 50 years of age”. This section now reads: Lines: 36-40 Known for its diverse risk factors, the chance of developing colorectal cancer increases markedly in patients older than 50 [3]. Notwithstanding, recent evidence suggests a steadily rising and alarming incidence and/or increasing burden of early-onset colorectal cancer (disease in patients younger than 50 years) in several countries, especially within the last three decades [1,4]. This makes screening from as early as 45 to 50 years an important part of care in most developed and some developing countries [3,4] Comment (3) The authors may want to consider moving the sentence "Staging on...specimen [24]" found in lines 93 to 95 after the sentence "In many cases... (FOBT) [24]" as this may potentially improve the flow of the manuscript. Response (3): The authors have reviewed the area identified and wish to agree with the reviewer that by rearranging the stated sentence will indeed improve the flow of the manuscript. See lines 101-102 The authors do wish to thank the reviewer for urging this response. Comment (4) Please consider adding a breakdown of colon and rectal cancer cases in Table 1. Response (4): The authors have taken careful note of the reviewer’s comments and wish to share that they have since included a breakdown of colon and rectal cancer cases in Table 1. The title of said table was also edited to represent the change. (see below) See lines 182 Table 2. Background attributes of the population of cases of colorectal cancer (combined and disaggregated) (2017-2021).
|
|||||||||||||||||||||||||||||||||||||||||||||||||||||||||||||||||||||
|
|||||||||||||||||||||||||||||||||||||||||||||||||||||||||||||||||||||
|
|||||||||||||||||||||||||||||||||||||||||||||||||||||||||||||||||||||
Kindly note that in addition to the edits done in respect of the comments and/or suggestions of the Reviewer, the authors have made some edits to further improve the article and so as to ensure that there is consistency across all areas of our study re: scope and/or purpose. This included edits to text, figures and tables. |
|||||||||||||||||||||||||||||||||||||||||||||||||||||||||||||||||||||
|
|||||||||||||||||||||||||||||||||||||||||||||||||||||||||||||||||||||
Thank you |

Reviewer 2 Report
Comments and Suggestions for Authors
The article analyzes the direct medical costs of colorectal cancer in Antigua and Barbuda, using a prevalence-based cost-of-illness methodology. The study is well-structured, and the methodology is described in detail. The results provide valuable insights for health policy development and resource allocation. However, there are several key aspects that require improvement. Details in the review file.

There are several key aspects that require improvement. Details in the review file.
Author Response
Response to Reviewer 2 Comments
|
||||||
1. Summary |
|
|
||||
Thank you very much for taking the time to review this manuscript. We do express our appreciations to you for your comments and suggestions offered. It is our hope that the revised manuscript has addressed your concerns. We do look forward to hearing from you on this. Please find the detailed responses below and the corresponding revisions/corrections highlighted in track changes in the re-submitted files.
|
||||||
2. Point-by-point response to Comments and Suggestions for Authors
|
||||||
The article analyzes the direct medical costs of colorectal cancer in Antigua and Barbuda, using a prevalence-based cost-of-illness methodology. The study is well structured, and the methodology is described in detail. The results provide valuable insights for health policy development and resource allocation. However, there are several key aspects that require improvement.
Comment (1) The introduction provides a solid background for the study, but it lacks a more detailed explanation of why this specific methodology was chosen and its limitations. Additionally, more discussion on the unique healthcare system challenges in Antigua and Barbuda in an economic context would be beneficial.
Response (1) The authors have taken a keen note of the reviewer’s comment and have inserted in the introduction section the following:
See lines 52-55: The prevalence-based methodology is commonly considered in costing studies due to its advantage in computing cost estimates attributable to a disease over a specified period, usually a year [10]. This can be achieved either retrospectively or prospectively, with key drawbacks being that it does not quantify the long-term consequences of the condition being measured, nor does it measure potential savings resulting from certain health interventions [10,14].
On the matter of “more discussion on the unique healthcare system challenges in Antigua and Barbuda in an economic context would be beneficial”, the authors have since inserted in the ‘Introduction’ in brief, a note to this effect.
See lines: a lone tertiary hospital that caters to most cancer cases diagnosed in the country [16].
And lines 61-63: As a country whose main contributor to GDP per capita is tourism [21], Antigua and Barbuda’s monetary policy and currency are guided by its relations with the Eastern Caribbean Central Bank and Eastern Caribbean Currency Union, respectively [22,23]. Moreover, for a country with a relatively high Human Capital Index [24],
Comment (2) The literature review is well-developed, but references to studies from comparable countries would provide a broader context. More information about alternative methods used in cost-of-illness analysis would also enhance this section. Response (2) The authors have taken a keen note of the reviewer’s comment and have since included a short sentence in the introduction section to reflect this. See lines 45-48 Several studies have estimated the economic burden of colorectal cancer using various cost-of-illness approaches, with most adopting either the incidence- or prevalence-based methods [9,10]. More specifically, several researchers have opted to estimate the direct medical costs of colorectal cancer in various settings using the prevalence-based cost-of-illness methodology [9].
And see lines 52-55:
The prevalence-based methodology is commonly considered in costing studies due to its advantage in computing cost estimates attributable to a disease over a specified period, usually a year [10]. This can be achieved either retrospectively or prospectively, with key drawbacks being that it does not quantify the long-term consequences of the condition being measured, nor does it measure potential savings resulting from certain health interventions [10,14].
Comment (3) The research methodology is precise, but there is a lack of explanation regarding potential sources of error in data analysis. For instance, were possible inaccuracies in medical documentation considered? Moreover, the retrospective nature of the study could lead to an underestimation of costs.
Response (3) The authors have taken note of the reviewer’s comment and wish to share that we had sought to address ‘possible inaccuracies in medical documentation considered’ and any potential for an ‘underestimation of costs’ in our analyses by including a section in our methods titled “Sensitivity analysis.”
Under this subsection in the methods section of the paper, we included the following:
“Applying an approach previously adopted and discussed in other studies, we conducted a sensitivity analysis to account for uncertainties in the derived cost estimates or possible inaccuracies in the data analysis [10,16,36,43]. Sensitivity analysis was also performed to account for any unrecorded cases at the study sites involved in our research. This was achieved by varying our cost estimates by a range of ± 25% [10,16,36,43].”
The authors also edited this section include the words ‘possible inaccuracies in data analysis,’ so as to lend clarity to the effect of the sensitivity analysis done on the reported study estimates. See lines
Comment (4) The results are clearly presented, but some tables and figures require better readability.
Response (4) The authors have since edited a section of Tables 3, 4 and 5 and as per editors suggestion. The section headed follow-care was changed to ongoing care and the line items consolidated to give three-line items, namely
Comment (5) Additionally, a more detailed discussion on the implications of the results for patients and doctors, as well as potential cost reduction strategies, would be valuable.
Response (5) The authors have taken note of this comment from the reviewer and wish to share that we have since addressed this point by making the following insertion into the discussion section.
See lines 288-294: Initiatives could focus on (i) engendering greater health-seeking behavior through various disease-sensitization campaigns that, among other things, encourage people to adopt healthy lifestyle behaviors, including adjustments to diet, engaging in regular physical activity, and seeking timely colorectal cancer screening [56]. These initiatives could also focus on (ii) encouraging clinicians to keep abreast of improvements in colorectal cancer management practices through regular participation in tumor board meetings and other medical education fora [57]. Initiatives could also be directed at reviewing the current approach to colorectal cancer care locally, in order to optimize the quality of care and thereby reduce the economic costs of managing this disease [56].
And lines 299-300:
including replacing expensive chemotherapeutic agents with equally therapeutic or more efficacious low-cost alternatives;
Comment (6) The discussion is well-structured, but the study limitations section should be expanded. For example, indirect costs such as productivity losses were not considered. The conclusions align with the results, but they could be formulated more precisely.
Response (6) The authors have taken note of the reviewer’s suggestions and wish to share that the absence of indirect costs and productivity losses were accounted for in our mention of the study’s limitations. This is stated in:
See lines
This meant that cost elements such as end-of-life care, productivity losses, and other indirect medical costs were not considered in our cost estimates [58]
Additionally, and on the matter of “The conclusions align with the results, but they could be formulated more precisely.”, kindly note that the authors, while not changing this section much, have since revised it to read thus:
See lines 329-339 Our study presented up-to-date evidence regarding the economic burden of colorectal cancer in Antigua and Barbuda. The estimates of annual direct medical costs appear to be substantial given the local context, with the major drivers being treatment (54%) and post-treatment side-effects care (35%). Employing cost containment measures, such as revising drug procurement practices, could immensely reduce these costs going forward. Optimizing data collection via a national cancer registry and register of costs, as well as engaging in studies using cost-effectiveness and cost-utility models could allow for more extensive cost estimates in the future. Moreover, notwithstanding the usefulness of our cost estimates to health administrators for (i) designing new and/or improving existing policies, (ii) assisting in budget planning, and (iii) implementing guidelines for allocating resources used in colorectal cancer care locally, these health management parameters could be better informed should future cost-of-illness studies consider productivity losses and other elements of indirect costs in assessing the economic burden of colorectal cancer in Antigua and Barbuda.
Comment (7) The quality of English in the article is generally good, but there are some grammatical and syntactic errors that may affect readability. Specifically, attention should be given to: 1. Incorrect use of commas and sentence structures that sometimes disrupt the flow of reading. 2. Some medical and economic terms need to be used more precisely to avoid ambiguity. 3. Inconsistent verb tenses in different sections, which may lead to misinterpretation of results. 4. Lack of terminological consistency in some sections – for example, using different terms for the same concepts. 5. A professional language review would improve the clarity and formality of the text.
Response (7) The authors have taken keen note of the reviewer’s comments and wish to share that we have since subjected the revised manuscript to professional language editing to improve on the clarity and formality of the text. We have included a copy of the language editing certificate as proof of us undertaking this particular exercise.
In summary, the article is a valuable contribution to the literature on the economic burden of colorectal cancer, but it requires revisions before publication. After these changes, it could serve as an important resource for policymakers and researchers.
|
||||||
|
||||||
|
||||||
Kindly note that in addition to the edits done in respect of the comments and/or suggestions of the Reviewer, the authors have made some edits to further improve the article and so as to ensure that there is consistency across all areas of our study re: scope and/or purpose. This included edits to text, figures and tables. |
||||||
|
||||||
Thank you |

Reviewer 3 Report
Comments and Suggestions for Authors
Article: Analysis of the Direct Medical Cost of Colorectal Cancer in Antigua and Barbuda: A Prevalence -Based Cost of illness Study
Thank you for submitting your manuscript to this journal. Below are a few suggestions for improvement:
Abstract
The abstract is well presented.
1. Introduction
The introduction is inadequate and could benefit from further development. It would be helpful to include key points, a clear outline of the study's objectives, and present some literature related to the study. In my opinion, this would make it easier for the reader to understand the authors' intentions.
2. Materials and methods
It should also explain the types of data, their sources, and how they are organized. Lines 70-80 could be expanded a bit more to do this.
2.2- lines 88–103, could be improved with some additional information to enhance clarity and make it easier for the reader to follow. The content itself is satisfactory
Suggestion: After reviewing Figure 1 and the tables, it appears that including a table displaying the average costs or range of average costs/expenses for each stage of the screening process, along with other direct costs, could provide clearer insights and be more helpful for the reader.
2.3 – reader would like to see the exact based year authors used in this study for CPI (Lines 134- 138)
2.4 – A bit explanation would do better here how the sensitive analysis was done for your study.
3. Results
Overall, it is presented well.
Section 3.2, lines 177–183, needs a clearer explanation of how the single-year figure was calculated. Given that the study period spans 2017–2021 (15), it is unclear how the number 11 was derived out of 22. Also, the reasoning behind the 50% increase?
4. Results
Overall OK.
5. Conclusion
Including a paragraph discussing the limitations of the study and providing suggestions for future research would enhance the overall analysis. For instance, while the study adopts a prevalence-based cost-of-illness approach, it focuses solely on direct costs, leaving out associated indirect costs. Highlighting this limitation and noting that indirect costs should be considered, especially when informing health cost policies, would be valuable.
References:
The reference section could be enhanced by including more recent research and findings.
Author Response
Response to Reviewer 3 Comments
|
|||||
1. Summary |
|
|
|||
Thank you very much for taking the time to review this manuscript. We do express our appreciations to you for your comments and suggestions offered. It is our hope that the revised manuscript has addressed your concerns. We do look forward to hearing from you on this. Please find the detailed responses below and the corresponding revisions/corrections highlighted in track changes in the re-submitted files.
|
|||||
2. Point-by-point response to Comments and Suggestions for Authors
|
|||||
Comments and Suggestions for Authors Article: Analysis of the Direct Medical Cost of Colorectal Cancer in Antigua and Barbuda: A Prevalence -Based Cost of illness Study
Comment (1)
Response (1) The authors do humbly acknowledge the reviewer’s remarks
Comment (2)
Response (2) The authors have taken note of the reviewer’s comments and wish to share that consistent with the comments of the first and second reviewers, we have made some additions to this section so as to improve readability.
See lines 45-48 Several studies have estimated the economic burden of colorectal cancer using various cost-of-illness approaches, with most adopting either the incidence- or prevalence-based methods [9,10]. More specifically, several researchers have opted to estimate the direct medical costs of colorectal cancer in various settings using the prevalence-based cost-of-illness methodology [9].
Lines 56-63 Antigua and Barbuda is a country in the English-speaking Leeward Islands with a projected 2021 population of 99,337 [15], with a lone tertiary hospital that caters to most cancer cases diagnosed in the country [16]. Public healthcare is mostly financed through statutory deductions [17,18]. Colorectal cancer is ranked among the top five causes of cancer deaths locally [19], and its incidence reflects the general trends observed elsewhere in the Caribbean [20]. As such, there is a burgeoning need to understand this cancer’s economic impact. As a country whose main contributor to GDP per capita is tourism [21], Antigua and Barbuda’s monetary policy and currency are guided by its relations with the Eastern Caribbean Central Bank and Eastern Caribbean Currency Union, respectively [22,23]. Moreover, for a country with a relatively high Human Capital Index [24],
The authors do hope that through these edits that we made have addressed the reviewer’s comments.
Comment (3) 2. Materials and methods
Response (3) The authors have taken note of the reviewer’s comments and wish to share that concerning the data types and their sources, and how they are organized, we would have articulated this in the following subsections
· 2.2 Table 1 which itemizes the data categories, variables and data sources of cost information.
· 2.3 which addresses sources of both patient data and cost data
· 2.5 which shared in brief and as part of ethical requirement, when the types of data were collected.
Additionally, and so as not to lose the significance of the authors suggestion and improve clarity in respect of patient data sources, we have since replaced the word ‘taken’ found in subsection 2.1 with the word abstracted since this speaks to both the issue of source and process of collection of patient data.
See line 81
Response (4) The authors have reviewed this section and have inserted the following lines into the text.
Lines 98-100 If the biopsy comes back positive for malignant cells, then staging is performed to ascertain the extent of the disease and to decide on the appropriate care strategy or treatment plan [33].
And lines 109-110 The costs related to all aspects of colorectal cancer management based on 2021 market prices were considered for this study, as derived from the sources highlighted in Table 1 [16].
Comment (5) Suggestion: After reviewing Figure 1 and the tables, it appears that including a table displaying the average costs or range of average costs/expenses for each stage of the screening process, along with other direct costs, could provide clearer insights and be more helpful for the reader.
Response (5)
The authors wish to thank the reviewer for making this observation. To this end we have since inserted another table in the manuscript. This captures the range of average costs for each stage based on the broad care parameters identified as per the other tables included in the manuscript.
See line 212
Table 3B: Range of costs for staging, management and treatment of colorectal cancer stages I-IV
Additionally, and in this regard, we reviewed our cost data obtained from archived reimbursement records and had cause to make an adjustment to correct an inadvertent understatement of the cost of chemotherapy port insertion in our tables.
Response (6) The authors wish to recognize the reviewer’s comments in respect of the need to show the exact based year used for CPI. The authors wish to share that this information is articulated in lines of the manuscript.
Lines
CPI in 2021 = 95.27; CPI in based year = 95.27
Its usage is further explained by the example: The cost of a colonoscopy in 2021 based on reimbursement records seen was XCD 3500.00 Converted to USD: 3500.00 / 2.7169 = USD 1288.23 the base year price Value in 2021 USD = 1288.23 x (CPI in 2021/ CPI in base year) This gives Value in 2021 USD = 1288.23 x (95.27 / 95.27) = USD 1288.23
Therefore, for ease of understanding, if an estimated 2 persons diagnosed with early-stage cancer had a colonoscopy in a year, this would in effect be used to give the direct medical costs for colonoscopy in a single year as: 2 x 1288.23 = USD 2576.46 The authors do hope that we have addressed the reviewer’s comment satisfactorily.
Response (7 ) So as to lend clarity to how sensitivity analysis was done, the authors have revised this section to read:
Lines 162-165 Applying an approach previously adopted and discussed in other studies, we conducted a sensitivity analysis to account for uncertainties in the derived cost estimates or possible inaccuracies in the data analysis [10,16,36,43]. Sensitivity analysis was also performed to account for any unrecorded cases at the study sites involved in our research. This was achieved by varying our cost estimates by a range of ± 25% [10,16,36,43]. The authors do thank the reviewer for urging this response.
Comment (8 ) 3. Results
Response ( 8) The authors do humbly acknowledge the reviewer’s remarks
Given that the study period spans 2017–2021 (15), it is unclear how the number 11 was derived out of 22. Also, the reasoning behind the 50% increase?
Response ( 9) The authors have taken a keen note of the reviewer’s comments and after much deliberation thought it best to provide for readers a simple equation which we believe would lend ease of understanding on how the average number of cases per single year was determined. This has since been inserted in the subsection 3.2 of the results section.
See lines 188-195: This is further explained by the equation below: Patients in a single year (Cal) = Cc is the number of diagnosed cases of colorectal cancer (2017-2021) = 79; Da is the number of patients with colorectal cancer who were diagnosed between 2017 and 2021 and died in the same period = 22; Cal is the average prevalent cases (patients in a single year) = 11. Moreover, when the average number of diagnosed cases in a single year increased by 50%, the estimated cases per year became Cal = 17 Comment (10) Also, the reasoning behind the 50% increase?
Response (10) The authors have taken note of the reviewer’s comments and wish to share that our rationale for using the ‘50%’ increase was done to further ascertain or explore how any unrecorded cases by the study sites could influence our derived cost estimates. We adopted this position based on two important points
a. The fact that Antigua and Barbuda lacks an established national cancer registry or national database on cancer cases. b. The country lacked an active or systematic screening programme for colorectal cancer during the period 2017-2021.
Reference: International Agency for Research on Cancer. CanScreen 5; Country fact sheet: Antigua and Barbuda, Colorectal Cancer Screening Programme. World Heal Organ 2023. https://canscreen5.iarc.fr/?page=countryfactsheetcrc&q=ATG&rc= (accessed December 7, 2024).
The authors are of the view that by varying our estimated case count we can also allow for comparisons or usefulness of costs estimates to be made with and in other populations where there are ethnic, demographic and economic similarities as is the case in several other Caribbean islands.
The authors have since inserted a few words to this effect in the discussion section of the text.
See lines 326: while also useful for comparisons in other similar settings,
Additionally, the authors posit that a similar approach on varying the number of cases was used by Ginindza and colleagues (Section Materials & Methods, subsection Cervical cancer, paragraph 3: Ginindza TG, Sartorius B, Dlamini X, Östensson E. Cost analysis of Human Papillomavirus-related cervical diseases and genital warts in Swaziland. PLoS One 2017;12:e0177762. https://doi.org/10.1371/journal.pone.0177762.
Comment (11) 4. Results
Response (11) The authors do humbly acknowledge the reviewer’s remarks
Comment (12) 5. Conclusion
Response (12) The authors have taken a keen note of the reviewer’s comments and consistent with comments made by the second reviewer as well, have since made a few edits to the conclusion section of the manuscript to cater to some of the points raised.
See lines 329-339 Our study presented up-to-date evidence regarding the economic burden of colorectal cancer in Antigua and Barbuda. The estimates of annual direct medical costs appear to be substantial given the local context, with the major drivers being treatment (54%) and post-treatment side-effects care (35%). Employing cost containment measures, such as revising drug procurement practices, could immensely reduce these costs going forward. Optimizing data collection via a national cancer registry and register of costs, as well as engaging in studies using cost-effectiveness and cost-utility models could allow for more extensive cost estimates in the future. Moreover, notwithstanding the usefulness of our cost estimates to health administrators for (i) designing new and/or improving existing policies, (ii) assisting in budget planning, and (iii) implementing guidelines for allocating resources used in colorectal cancer care locally, these health management parameters could be better informed should future cost-of-illness studies consider productivity losses and other elements of indirect costs in assessing the economic burden of colorectal cancer in Antigua and Barbuda.
Response (13) The authors have taken note of the reviewer’s comments and wish to share that our reference section has been enhanced with the inclusion of a few more important references that were consulted to help us in presenting our research work.
|
|||||
|
|||||
|
|||||
Kindly note that in addition to the edits done in respect of the comments and/or suggestions of the Reviewer, the authors have made some edits to further improve the article and so as to ensure that there is consistency across all areas of our study re: scope and/or purpose. This included edits to text, figures and tables. |
|||||
|
|||||
Thank you |

Reviewer 4 Report
Comments and Suggestions for Authors
My only concern with the manuscript is with Table 3, where the values of the unit cost column are repeated exactly in all columns for the different stages and number of cases. In spite of this, the totals at the bottom of the columns differ, obviously giving the sum of other values than those above.
Also, the text 'Figure x ....' should appear below the figure, not above
Author Response
Response to Reviewer 4 Comments
|
||
1. Summary |
|
|
Thank you very much for taking the time to review this manuscript. We do express our appreciations to you for your comments and suggestions offered. It is our hope that the revised manuscript has addressed your concerns. We do look forward to hearing from you on this. Please find the detailed responses below and the corresponding revisions/corrections highlighted in track changes in the re-submitted files.
|
||
2. Point-by-point response to Comments and Suggestions for Authors
|
||
Comment (1) My only concern with the manuscript is with Table 3, where the values of the unit cost column are repeated exactly in all columns for the different stages and number of cases. In spite of this, the totals at the bottom of the columns differ, obviously giving the sum of other values than those above. Response (1) The authors have taken a keen note of the reviewer’s comment and wish to share that consistent with a suggestion of Reviewer #3, we have since created a table that further summarizes the direct medical unit costs of colorectal cancer by clinical stage of the disease. As per the reviewer’s suggestion, this table titled, Table 3B, seeks to express the direct medical unit cost for each stage by giving a summary of costs according to each care parameter while also presenting each summary cost by way of a range of ± 25%. Note that the varying of costs by ± 25% is consistent with the approach outlined in our methods, subsection 2.4 on sensitivity analysis. Additionally, kindly note that the differing of totals per each stage is consistent with parameters involved in the general management of each disease stage, which in turn is consistent with the guidance gained from experts in the field. Comment (2) Also, the text 'Figure x ....' should appear below the figure, not above Response (2) The authors have taken note of the reviewer’s comment and do wish to share that we have since relocated the text identifying each figure to a place immediately below said figures.
|
||
|
||
|
||
Kindly note that in addition to the edits done in respect of the comments and/or suggestions of the Reviewer, the authors have made some edits to further improve the article and so as to ensure that there is consistency across all areas of our study re: scope and/or purpose. This included edits to text, figures and tables. |
||
|
||
Thank you |

Round 2
Reviewer 1 Report
Comments and Suggestions for Authors
No further comments on this manuscript. The reviewers did a great job addressing the previous comments.
Reviewer 2 Report
Comments and Suggestions for Authors
The authors made the changes requested in the review.